# Biological Treatments in Inflammatory Bowel Disease: A Complex Mix of Mechanisms and Actions

Lorena Ortega Moreno [1,2,3], Samuel Fernández-Tomé [1,3] and Raquel Abalo [4,5,6,7,*]

1 Instituto de Investigación Sanitaria Hospital Universitario de La Princesa, 28006 Madrid, Spain; lorena.ortega8317@gmail.com (L.O.M.); fernandeztome.samuel@gmail.com (S.F.-T.)
2 Department of Medicine, Faculty of Medicine, Universidad Autónoma de Madrid, 28049 Madrid, Spain
3 Centro de Investigación Biomédica en Red de Enfermedades Hepáticas y Digestivas (CIBERehd), 28029 Madrid, Spain
4 Department of Basic Health Sciences, Faculty of Health Sciences, University Rey Juan Carlos (URJC), 28922 Alcorcón, Spain
5 High Performance Research Group in Physiopathology and Pharmacology of the Digestive System (NeuGut-URJC), URJC, 28922 Alcorcón, Spain
6 Associated I+D+i Unit to the Institute of Medicinal Chemistry (IQM), Scientific Research Superior Council (CSIC), 28006 Madrid, Spain
7 Working Group of Basic Sciences in Pain and Analgesia of the Spanish Pain Society (Grupo de Trabajo de Ciencias Básicas en Dolor y Analgesia de la Sociedad Española del Dolor), 28046 Madrid, Spain
* Correspondence: raquel.abalo@urjc.es; Tel.: +34-91-488-8854

**Abstract:** Inflammatory bowel disease (IBD) is a chronic disease that requires lifelong medication and whose incidence is increasing over the world. There is currently no cure for IBD, and the current therapeutic objective is to control the inflammatory process. Approximately one third of treated patients do not respond to treatment and refractoriness to treatment is common. Therefore, pharmacological treatments, such as monoclonal antibodies, are urgently needed, and new treatment guidelines are regularly published. Due to the extremely important current role of biologics in the therapy of IBD, herein we have briefly reviewed the main biological treatments currently available. In addition, we have focused on the mechanisms of action of the most relevant groups of biological agents in IBD therapy, which are not completely clear but are undoubtfully important for understanding both their therapeutic efficacy and the adverse side effects they may have. Further studies are necessary to better understand the action mechanism of these drugs, which will in turn help us to understand how to improve their efficacy and safety. These studies will hopefully pave the path for a personalized medicine.

**Keywords:** action mechanism; anti-integrins; anti-TNF-α; biologics; Crohn's disease; inflammatory bowel disease; monoclonal antibodies; ulcerative colitis



## 1. Introduction

Inflammatory bowel disease (IBD) is a chronic disease characterized by intestinal inflammation with a relapsing and remitting clinical course that generally requires lifelong medication and is associated with significant morbidity, hospitalization needs, and productivity losses [1]. Furthermore, the disease is progressive, with damage accumulation and treatment failure over time. Additionally, IBD is considered a systemic disease, with extra-intestinal manifestations and symptoms frequently affecting the joints, skin, eyes, and (although less often) the liver, pancreas, or lungs, which can also contribute to morbidity and reduced quality of life [2–4]. Importantly, incidences of the disease are increasing world-wide. The highest rates have been traditionally found in North America and Europe, but currently there is a worrying trend of increasing occurrence of the disease in previously low-incidence regions (e.g., Asia, South America . . . ), which is likely associated with adoption of a westernized mode of life involving varied factors such as diet pollution, microbial

exposure, sanitation [5–7], and possibly even psychological stress [8,9]. In addition, IBD is mainly diagnosed at a young age, so its prevalence is also high (1.6 million persons in US and 2.2 million persons in Europe) [10,11]. Not surprisingly, the costs associated with this disease are also high (1.7 million dollars per year in US and more than 2.5 million euro in Europe).

There are two main subtypes of IBD: Crohn's disease (CD) and ulcerative colitis (UC). These subtypes have different clinical presentation and histopathological findings [1]. However, some features are shared by both IBD types including clinical features (loss of weight and appetite, rectal bleeding, diarrhea, tenesmus, anemia), endoscopic features (erythema, loss of vascular pattern, erosions/ulcerations, spontaneous bleeding) and pathological features (crypt architecture distortion, higher in UC than in CD; crypt abscesses and shortening; infiltration of leukocytes into lamina propria) [1].

IBD is characterized by impairment of the epithelial and mucus layer barrier via disruption of tight junctions and inflamed lamina propria. This is associated with dysbiosis (altered gut microbiome composition), whose role as a causative factor or a consequence of mucosal inflammation is not yet clear [12,13]. Furthermore, the mucosal immune system constitutes the third most recognized component contributing to the complex underlying etiopathogenic mechanisms [14]. Indeed, pronounced infiltration of the lamina propria with a mix of neutrophils, macrophages, dendritic cells, and natural killer (NK) T cells is found in active IBD [15]. Increased Th1, Th2, Th9, Th17 and Th17.1 responses, as well as reduced Treg and Tr1 responses, have all been suggested to play a role in IBD pathophysiology, although it is highly unlikely that all of these responses are altered in each individual patient [16]. Thus, currently, the most accepted etiopathogenic theory is that IBD is caused by an impairment in immunological tolerance, resulting in an exacerbated immune reaction against intestinal microbiota in genetically susceptible individuals and thereby facilitating mucosal inflammation [13].

Biomarkers common to both UC and CD are fecal calprotectin (useful in screening IBD for endoscopic evaluation and clinical management of IBD) and fecal lactoferrin (used for assessing the course of disease activity and healing). These two intestinal inflammatory conditions share many genetic and environmental risk factors [1]. For example, it is recognized that antibiotics intake increases the risk of IBD, that psychological distress and sleep deprivation correlate with flare-ups, that depression and anxiety cause clinical recurrence, and that animal-based diet is harmful [1], although other contributing factors are more disease-specific [1,6].

Importantly, both types of IBD have been associated with an increased risk of developing colorectal cancer (CRC), primarily associated with the occurrence of chronic intestinal inflammation and extra-intestinal malignancies, which are related with both the chronic use of immunosuppressive therapies and an underlying inflammatory state [17,18]. The risk of developing CRC or extra-intestinal cancer increases with time since diagnosis (for example, the risk of developing CRC is high after six–eight years and increases linearly year by year) and the extension of inflammation [17–19]. However, recent studies [20] have presented robust data showing that this risk may not be as high as initially reported (i.e., for CRC it is now considered to be about two-fold), which might be attributed to different factors such as better screening strategies and colectomy implementation for high-grade dysplasia, on the one hand, and the potent immunosuppressive and/or chemopreventive properties of the drugs currently used, on the other [17]. However, as mentioned, immunosuppressive treatment may induce important side effects, including extra-intestinal cancer. Immunosuppressive agents may cause tumor formation through direct alteration of DNA, impairment of immune control of chronic infection by mutagenic viruses (Epstein-Barr virus, human papilloma virus), or a reduction of immunosurveillance of cancer or dysplastic cells [17]. Thus, in IBD patients, both too much inflammation and too much immunosuppression may be harmful, and these patients need to be carefully monitored to maintain the right balance among the two factors, through selection of the right treatment at each stage of the disease [17].

As a matter of fact, there is currently no cure for IBD, and the therapeutic objective is to control the inflammatory process. This is not easy, since multiple inflammatory pathways are concurrently activated in the intestinal mucosa and the pathogenic mechanisms sustaining inflammation in IBD are dynamic and change over time. Accordingly, treatment of patients needs to take into account the symptoms, inflammatory status and mechanism of action of the drug/s with most likely beneficial impact to adequately control the disease at each particular moment. Despite all of these efforts, approximately one third of treated patients do not respond to treatment (the proportion of primary non-responders may be as high as 30–50%), refractoriness to treatment is common (10% of patients treated with biologics become refractory) and safety issues (development of infectious, neoplastic or, other side effects) are also a major concern for both patients and clinicians [17,21,22]. Therefore, new pharmacological and non-pharmacological therapies [21,23–26], as well as optimization of the currently available therapeutic strategies [27] are urgently needed, and new treatment guidelines are regularly published [28,29].

Traditional treatments for IBD, such as aminosalicylates (sulfasalazine, mesalazine), corticosteroids (budesonide, prednisone), and some immunomodulators (thiopurines, i.e., azathioprine and 6-mercaptopurine; methotrexate), were introduced several decades ago (since the 1950s) and are still main-stream therapies [1,30]. These drugs have several advantages such as their relatively small size (<1000 Da), stable structure, reduced production cost, short half-life, (which is an advantage in cases where rapid elimination is needed), and oral route of administration [1]. Although they provide symptom improvement, they may also cause relevant adverse effects (including carcinogenesis, particularly thiopurines) due to their broad immunosuppressive, antimetabolic, or unknown mode of action, and some patients are refractory to these treatments.

More targeted or specific pharmacologic treatments for IBD interfere with two main pathways (namely cytokine signaling and immune cell trafficking) and are classified into biologics (monoclonal antibodies) and small molecule drugs [25]. These drugs have revolutionized the treatment of IBD (particularly that of its severe forms), and new entities are being evaluated and even incorporated to clinical practice relatively quickly.

Biological therapies were introduced in the late 1990s to induce and maintain remission (i.e., infliximab was introduced for treatment of CD and UC in 1999 and 2006, respectively). These therapies use monoclonal antibodies targeting tumor necrosis factor-$\alpha$ (TNF-$\alpha$), integrins $\alpha$4, and cytokine molecules such as the common p40 subunit of IL-12 and IL-23 [31]. Monoclonal antibodies are expensive and need to be administered intravenously or subcutaneously since proteolytic gastrointestinal enzymes can destroy them [32]. Following parenteral administration, proteolytic catabolism eventually occurs after the internalization of the antibody by phagocytes of the reticuloendothelial system [33]. Nevertheless, monoclonal antibodies display a long half-life, which facilitates adherence to treatment but may also be a disadvantage in face of an infection, surgery, or pregnancy. One of the principal concerns with biologics is the fact that they can fail since the immune system may recognize them as foreign bodies and block their efficacy over time. Thus, although biological drugs have helped many patients to achieve remission, on many occasions they lose their efficacy. Moreover, no single marker can be used as a prognostic indicator of response to any biologic treatment in IBD [34]. Therefore, new biologics [25] and new combinations of different biological drugs are currently being studied as a possible means to increase efficacy and safety of these treatments [27]. In addition, other therapies, namely targeted small molecule drugs [25], may be useful.

Targeted small molecule drugs include Jak inhibitors, modulators of sphingosine-1-phosphate receptors (lymphocyte trappers), phosphodiesterase inhibitors, and oligonucleotide-based therapeutics [25]. As with the traditional IBD treatments mentioned above, these drugs are small chemical structures with a short half-life and a relatively low cost. These molecules have less potency and half-life than biologics, a generally less specific mechanism of action and, due to their broader diffusion (associated with their smaller size), a

greater risk of unspecified side effects. However, an important advantage is their lack of immunogenicity [35].

Due to the extremely important role of biologics in the current treatment of IBD, herein we have briefly reviewed the main biological treatments presently available for this increasingly impactful chronic disease. Some of the biologicals and biological-related therapies currently under clinical investigation have also been succinctly described. Finally, we have focused on the mechanisms of action of the most important monoclonal antibodies for IBD treatment, which are not completely clear but are undoubtfully key to understand both their therapeutic efficacy and adverse side effects.

## 2. Biological Therapies in IBD

As mentioned above, biological therapies use monoclonal antibodies. Monoclonal antibodies (mAbs) are immunoglobulins G (IgG), therapeutic proteins consisting of four polypeptide chains and two heavy and two light chains. There are two regions in the mAbs, the variable region (antigen-binding region, Fab) and the constant region (Fc). These mAbs are classified as murine antibodies with the suffix -omab; chimeric with the suffix -ximab; humanized with the suffix -zumab; and fully human with the suffix -umab [33] (Table 1).

**Table 1.** Main biologics (monoclonal antibodies) approved for IBD treatment.

| Type of Antibody | | Suffix | Anti-TNF-α | Anti-Integrin | Anti-Cytokine |
|---|---|---|---|---|---|
| | Murine | -omab | | | |
| | Human | -umab | Adalimumab (CD, UC) Golimumab (UC) | | Ustekinumab (CD, UC) |
| | Chimeric | -ximab | Infliximab (CD, UC) | | |
| | Humanized | -zumab | Certolizumab pegol (CD) | Natalizumab (CD) Vedolizumab (CD, UC) | |

Abbreviations: CD, Crohn's disease; IBD, inflammatory bowel disease; TNF, tumor necrosis factor; UC, ulcerative colitis. Created in BioRender.

### 2.1. Anti-TNF-α Therapy

#### 2.1.1. TNF-α

TNF-α is a pleiotropic cytokine involved in many biological activities, including cell proliferation, survival, and death. Although TNF-α is crucial for a normal immune response, when inappropriately or excessively produced it may be harmful and lead to diseases such as rheumatoid arthritis, psoriatic arthritis, psoriasis, noninfectious uveitis, and IBD, all of which are induced by the abnormal secretion of this cytokine. Thus, TNF-α has a key role in inflammation and the development and maintenance of chronic inflammatory diseases [36].

TNF-α is found in both a soluble and a transmembrane form. The transmembrane form is the initially synthetized precursor molecule and releases the soluble form after processing by the TNF-α converting enzyme (TACE), a membrane-bound disintegrin metalloproteinase. There are two receptors of TNF-α: TNFR1 (also termed TNFRSF1A, CD120a, and p55) and TNFR2 (also termed TNFRSF1B, CD120b, and p75). TNFR1 is expressed by all human tissues and is the key signaling receptor for TNF-α, whereas TNFR2 is generally expressed in immune cells and produces limited biological responses. Both soluble and transmembrane forms of TNF-α may activate TNFR1, but activities of the transmembrane form are relatively more TNFR2-dependent. Through complex intracellular pathways and molecular interactions (Table 2), TNF-α causes cytotoxic and proinflammatory responses via TNFR1 and facilitates cell activation, migration, or proliferation via TNFR2 [36].

**Table 2.** Main mechanisms of action involved in activation of TNF-α receptors and main biologic effects.

| | TNFR1 | | | TNFR2 |
|---|---|---|---|---|
| Alternative names | TNFRSF1A, CD120a, p55 | | | TNFRSF1B, CD120b, p75 |
| TNF-α form involved in activity | Soluble and transmembrane | | | Transmembrane |
| Intracellular transducer | TRADD | | | TRAF |
| Intracellular complexes | I | IIa and IIb (apoptosome) | IIc (necrosome) | I |
| Location of complex assemblage | Plasma membrane | Cytoplasm | Cytoplasm | Plasma membrane |
| Complex components | TNRF1, TRADD, RIPK1, TRAF (2 or 5), cIAP (1 or 2), LUBAC | TNRF1, TRADD, RIPK1, TRAF2, cIAP (1 or 2), pro-caspase-8, FADD (+ RIPK3 in complex IIb) | TNRF1, TRADD, RIPK (1 or 3) | TNFR2, TRAF2, TRAF2, cIAP1, cIAP2 |
| Final intracellular effector | NF-κB, MAPKs | Caspase-8 | MLKL | MAPKs, NF-κB, AKT |
| Biological effect | Inflammation, tissue degeneration, host defense, cell proliferation, cell survival | Apoptosis | Necroptosis, inflammation | Tissue regeneration, cell proliferation, cell survival, host defense, inflammation |

Abbreviations: AKT, protein kinase B; cIAP, cellular inhibitor of apoptosis protein; FADD, Fas-associated protein with death domain; LUBAC, linear ubiquitin chain assembly complex; MAPK, mitogen-activated protein kinases; MLKL, mixed lineage kinase domain-like protein; NF-κB, nuclear factor κB; RIPK, receptor-interacting serine/threonine-protein kinase; TNF, tumor necrosis factor; TNFR1 and TNFR2, TNF-α receptor types 1 and 2; TRADD, TNFR1-associated death domain; TRAF, TNFR-associated factor.

TNF-α plays a key role in the immune-mediated pathogenesis of IBD [37]. Indeed, in IBD the production of soluble and membrane-bound TNF-α in the intestinal mucosa is significantly increased in CD14+ macrophages, fibroblasts, and T cells [38]. On their target cells (macrophages and others), TNF-α induces the activation of nuclear factor kappa-B (NF-κB) that stimulates cell proliferation and induces the release of cytokines such as interleukin (IL)-1β and IL-6, as well as C-reactive protein, all of which play a role in inflammation [39]. These molecules cause the accumulation of different cell types in the lamina propria of the intestinal mucosa including fibroblasts, neurotrophils, and macrophages. Fibroblasts will cause fibrosis and stricture formation. Elastase secreted by neutrophils will cause intestinal matrix degradation. Macrophages, through the production of more inflammatory cytokines, will further increase matrix degradation, epithelial damage, endothelial activation, and vascular disruption [36].

2.1.2. Anti-TNF-α Antibodies in Current IBD Therapy

Different anti-TNF-α antibodies have been developed and used for the treatment of IBD and other immune-mediated inflammatory diseases since 1998 [36]. Four of them are widely used in the treatment of IBD: infliximab, adalimumab, golimumab, and certolizumab pegol (Table 1).

Anti-TNF-α antibodies block soluble TNF-α, thus preventing pro-inflammatory signal transduction, leading to the apoptosis of T-cells [39] and the production of anti-inflammatory cytokines [40]. In general terms, it is assumed that antibodies against TNF-α inactivate this pro-inflammatory cytokine by direct neutralization [41]. Inhibition of the membrane-bound TNF/TNFR2 pathway is thus the basis to induce T-cell apoptosis [42] and the consequent inhibition of downstream pro-inflammatory signals. Nonetheless, and considering the complexity of TNF-α signaling, it is generally accepted that anti-TNF-α antibodies may display more complex effects in addition to the simple TNF blockade [43,44], as discussed below. Moreover, the affinity of the different antibodies to TNF-α and their

cross-linking towards membrane-bound TNF-$\alpha$ has been found to be unequal between these drugs in several bioassays [45,46].

Etanercept, a fusion protein with two identical TNFR2 extracellular regions connected to the Fc fragment of human IgG1 [47], was the first anti-TNF-$\alpha$ agent approved by the Food and Drug Administration (FDA) for treatment of rheumatoid arthritis [36]. However, it is not used in IBD. In fact, in contrast to other common anti-TNF-$\alpha$ antibodies such as infliximab and adalimumab, etanercept binding to circulating TNF-$\alpha$ did not show efficacy in CD, thereby suggesting that other mechanisms beyond TNF-$\alpha$ neutralization are involved in the therapeutic effect of anti-TNF-$\alpha$ in IBD [48]. Indeed, some studies have hypothesized that both membrane-bound and soluble TNF have to be neutralized to induce T-cell apoptosis [49,50], but this has not been fully demonstrated.

Infliximab, a recombinant chimeric IgG1 mAb, is produced by cell culture from Chinese hamster ovary cells [33], and it was the first monoclonal antibody approved for the treatment of patients with IBD. It was later used also for rheumatoid arthritis, ankylosing spondylitis, psoriatic arthritis, and plaque psoriasis. It binds both soluble and transmembrane forms of TNF-$\alpha$ and causes cell lysis of macrophages and T cells [51]. Infliximab induces and maintains clinical remission and mucosal healing in patients with IBD that are refractory or unresponsive to non-biologics treatment, and even works in the treatment of perianal fistulas [52,53]. The half-life of this antibody is approximately eight-ten days. The standard dose is 5 mg/kg administered intravenously with a phase of induction of zero, two, and six weeks and a maintenance treatment every eight weeks. The response to this therapy is different depending on the person [32]. This might be at least partly due to the fact that there is a huge variability in the pharmacokinetics of infliximab among patients. Furthermore, some patients lose response over time [32]. Primary non-responders are those who do not have clinical response during the phase of induction while secondary non-responders are those who lose response to therapy over time. The factors that may increase the risk of being primary non-responders are high body mass index, smoking, high IL-8 level, and genetic mutations in the apoptosis-related genes. On the other hand, secondary refractoriness may occur when there is not enough active drug available, where immunogenicity may play a key role [54].

Adalimumab is a fully human recombinant IgG1 mAb that is produced, such as infliximab, from cultured Chinese hamster ovary cells. It blocks the binding of TNF-$\alpha$ to its receptors in an identical manner to the natural human IgG1. This leads to the lysis of cells with transmembrane TNF-$\alpha$ [51]. With a longer half-life than infliximab, approximately 10–13 days, it requires less frequent administration, which is subcutaneous.

Golimumab is a recombinant, completely humanized IgG1 mAb with higher affinity and neutralizing potency towards the soluble form of TNF-$\alpha$ compared with the trans-membrane form. It is beneficial for patients after the failure of infliximab and adalimumab therapy and has a half-life of seven to twenty days. It is safe and maintains its efficacy after two years of maintenance therapy [55,56].

Certolizumab pegol is a modified human mAb lacking the Fc region but carrying a polyethylene glycosylated (PEG) Fab fragment. It is produced by cell culture using *E. coli* [33,57] and, unlike infliximab and adalimumab, it does not induce apoptosis of immune cells due to the lack of an Fc region. Furthermore, it has more significant distribution into inflamed tissues compared to those antibodies. This also favors an increase in its half-life to two weeks. Results from some studies showed clinical benefits in patients refractory to other biological therapies [58]. Earlier studies showed more effectiveness when it is used as first-line agent [51].

The pharmacokinetic behavior of anti-TNF-$\alpha$ antibodies depends on the route of the drug administration. Intravenous therapy (infliximab) is less immunogenic and has a faster distribution than subcutaneous therapy (adalimumab and certolizumab). The mAbs clearance from the circulation is via proteolytic catabolism after the endocytosis in the reticuloendothelial system [59].

A high proportion of IBD patients treated with conventional anti-inflammatory agents need to change to mAbs treatments, and anti-TNF-α therapy has been key to reduce the rate of surgery in patients with IBD. However, this therapy is not without drawbacks. Thus, TNF-α antagonists have therapeutic efficacy but more than one-third of patients show no response to induction therapy and up to 50% of responders become non-responders over time [60]. Moreover, some patients have symptoms that do not correlate with elevated biomarkers of inflammation and it is necessary to undergo colonoscopy in those cases [61].

In contrast with thiopurines, which increase the risk of hematological malignancies, it is not currently clear if lymphoma risk is increased in IBD patients receiving anti-TNF-α antibodies. However, long-term accumulation could increase the risk, as reported in rheumatologic studies [17]. Other cancers whose development risk is also clearly increased with the use of thiopurines, such as urinary tract cancer, do not seem to be associated with anti-TNF-α therapies. However, robust data have been reported regarding the increased risk (1.5–4-fold) of melanoma development associated with anti-TNF-α treatment [20,62]. Importantly, this adds to a somehow intrinsic risk for melanoma in IBD that cannot be explained by the use of immunosuppressive agents, meaning that IBD patients, particularly if treated with anti-TNF-α antibodies (or with thiopurines), should be regularly evaluated by a dermatologist to detect possible early skin cancer lesions [63]. Furthermore, anti-TNF-α antibodies should be avoided in patients with a history of melanoma [63].

Finally, the use of TNF-α antibodies may cause tuberculosis, hepatitis B, herpes zoster, psoriasis, and other infections (due to immunosuppression), as well as cardiotoxicity, which is particularly worrying for elderly individuals [64]. Thus, other alternatives, with a better benefit-risk profile, are necessary [65].

## 2.2. Anti-Integrin Therapy

In case of primary failure, it is recommended that IBD treatment is switched to a molecule with a different mechanism of action. Anti-integrin drugs prevent the traffic of inflammatory cells that mediate the inflammatory process in IBD. These drugs are important for those IBD patients who do not respond to an anti-TNF-α treatment. There are two anti-integrins currently available in the clinics, namely natalizumab and vedolizumab (Table 1).

Integrin is a leukocyte heterodimeric transmembrane receptor formed by two subunits, α and β, and it is divided into several groups depending on the structure of these subunits. Different populations of leukocytes express different integrins. Thus, α4β1 is found in most leukocytes, α4β7 is present in gastrointestinal lymphocytes and αEβ7 is expressed in intraepithelial T cells, dendritic cells, and regulatory T cells [66]. These integrins bind to vascular endothelial cell adhesion molecule-1 (VCAM-1) and mucosal addressin cell adhesion molecule-1 (MadCAM-1) on endothelial cells and to E-cadherin on mucosal epithelial cells [67]. The increase of the expression of cell adhesion molecules (CAMs) provokes the immigration of leukocytes to the intestinal mucosa and the recruitment of immune cells to the inflammation site, which is essential for the maintenance of inflammation [65]. In addition, integrins and their ligands may participate in the pathogenesis of extra-intestinal inflammatory manifestations of IBD [68,69]. Anti-integrin therapy blocks the interaction of integrin on the surface of circulating immune cells with endothelial CAMs, preventing the intestinal recruitment of lymphocytes to the inflammation site [70].

The anti-integrin drug natalizumab is a recombinant humanized IgG4 that targets the α4 subunit of the integrins α4β1 and α4β7 on leukocytes. This drug stops the migration of inflammatory cells across the cell layers and needs to be administered for a long term to achieve positive results [51]. In the very beginning, this drug was approved by the FDA for multiple sclerosis treatment and later for CD, but it is only used in moderate to severe cases of CD due to its adverse effects [65], particularly progressive multifocal leukoencephalopathy (which is associated with the blockade of α4β1 integrin/VCAM-1 interactions by this drug in the central nervous system) [69,71].

Vedolizumab is a humanized mAb that inhibits the adhesion of leukocytes to the endothelium by blocking the α4β7 interaction with MadCAM-1 in the gastrointestinal tract. It is useful in cases of refractory IBD when corticosteroids or immune modulators have failed. Indeed, the FDA and European Medicine Agency (EMA) approved vedolizumab for the treatment of moderate to severe UC and CD patients that did not respond to the anti-TNF-α treatment. However, its efficacy may be greater in IBD patients naïve to anti-TNF-α therapy [72,73]. It is very specific and could push IBD patients to remission [51]. This therapy seems to be more effective than anti-TNF-α therapy in the maintenance phase [74] but requires longer treatment times to exert its full effect [69]. Importantly, it does not cause strong immunosuppressive systemic effects since it acts selectively in the intestine. However, due to its high selectivity, vedolizumab is not effective to reduce extra-intestinal symptoms. On the contrary, it may cause a migration of immune cells to organs other than the gut increasing the risk of extra-intestinal manifestations [65]. It has not been associated with an increased risk of serious or opportunistic infections and the risk of malignancy is not higher than baseline rates of patients with IBD [75]. Whether or not patients treated with vedolizumab are more susceptible to enteric pathogens such as cytomegalovirus, giardia, or *Clostridium difficile* is yet to be determined [69].

### 2.3. Anti-Cytokine Therapy

Ustekinumab is a fully humanized IgG1k mAb (Table 1) that binds the shared p40 subunit of cytokines IL12 and IL-23 preventing the binding of the cytokine to its receptor and reducing the activation of immune cells, thus reducing symptoms in active CD [51]. IL-12 consists of the heterodimer of p35 and p40 while IL-23 is made up of p19 and p40 subunits. In the presence of IL-12 and activated CD4+, T cell differentiates into a Th1 cell that increases interferon (IFN) γ production. IL-23 promotes the formation of Th17 cells [76]. The neutralization of IL-12 and IL-23 inhibits the cytokine production that is involved in the pathogenesis of CD, inducing remission in this disease [77,78]. The neutralization of IL-12 and IL-23 does not affect immune responses stimulated through other cytokines or cellular activities [77]. There is a precise specificity in the molecular interaction between ustekinumab and IL-12/23p40.

Ustekinumab shows clinical efficacy in psoriasis, psoriatic arthritis, and moderate to severe CD [76,79]. The incidence of the development of neutralizing antibodies is low and ustekinumab has a flexible dosage. The induction phase requires intravenous administration but during the maintenance phase the administration is subcutaneous, which is an advantage for the patient [80]. In most CD patients, remission is maintained after three years [81]. Furthermore, effectiveness of ustekinumab has also been demonstrated in UC [82]. This drug is now approved for both types of IBD [25].

### 2.4. New Biologics

New biologics or biologic-related therapies are currently under development in an attempt to overcome the drawbacks associated with the approved treatments [25].

For example, a new anti-TNF-α oral formulation (AVX-470) is being developed to achieve gut specificity which would increase patient safety as well as comfort. Interestingly, this is not a monoclonal, but a polyclonal anti-TNF-α antibody derived from cow colostrum with less than 1% of antibodies specific for this key cytokine. However, it is considered a promising strategy due to the known safety of bovine milk-derived IgA and the fact that the antibodies are released in the small intestine and colon [25].

Etrolizumab, is a humanized monoclonal anti-β7 antibody that blocks both α4β7 and αEβ7. αEβ7 controls the epithelial retention of homed lymphocytes in intestinal inflammation [83]. Etrolizumab may internalize β7 and in that manner, the integrin is inhibited on the cell surface [84]. This antibody has not been approved for IBD treatment yet seems to be effective to induce remission in both UC and CD [83,85].

Many new agents targeting other cytokines, particularly IL-12/23 and IL-17 (downstream effector of IL-23), are also under deep evaluation in clinical trials. So far, the selective

p19 inhibition through IL-23 (but not IL-12) has not proved to be advantageous in terms of its efficacy or safety [25]. Furthermore, inhibition of the IL-23 effector cytokine IL-17 aggravates the bowel inflammatory condition, possibly due to a role of IL-17 in epithelial barrier maintenance and regulation of gut colonization by segmented filamentous bacteria [86,87]. Thus, safety data on these options will be key to determine their right place (if any) in IBD treatment [25].

In addition to the mentioned monoclonal antibodies that inhibit α4 (natalizumab), β7 (etrolizumab) or both integrin subunits (vedolizumab), abrilumab (another anti-α4β7 monoclonal antibody), PF-00547659 (an anti-MadCAM-1 monoclonal antibody), and AJM300 (a small molecule integrin-α4 inhibitor) are being evaluated. The main advantage of these new adhesion inhibitors is their good safety profile, particularly for elderly and multi-morbid patients with malignancies in their history. However, broader studies are required to completely exclude possible relevant risks [25].

Other biologics inhibit IL-17, such as bimekizumab, or inhibit the p19 subunit of IL-23, such as mirikizumab, which reduces the activity of Th17 pathway. Bimekizumab inhibits IL-17A and IL-17F ligands, ixekizumab inhibits IL-17A ligand and brodalumab inhibits IL-17 receptor. These molecules have a safe profile and do not increase rates of infections or malignancy [27,88] but have not yet been approved for clinical use.

## 3. A Deep Insight into the Mechanisms of Action of Biologics in IBD

As mentioned above, the mechanisms underlying the biological therapies, particularly anti-TNF-α antibodies are far from clear. Here below we focus on the mechanisms of action of the main biologics.

### 3.1. Anti-TNF-α Antibodies

Table 3 summarizes some studies that have been carried out to evaluate the mechanistic effects of anti-TNF-α antibodies. Indeed, different modes of action of these drugs in IBD studies have been suggested, including induction of antibody-dependent cell-mediated cytotoxicity, complement-dependent cytotoxicity and apoptosis, modulation of cellular populations, proliferation or cell activation, regulation of immune factors and cytokines, as well as modulation of angiogenesis, and the intestinal barrier function [41,44]. These effects have been proved in both in vitro and animal models, but also using experimental human samples derived from healthy volunteers and IBD patients.

**Table 3.** Summary of studies evaluating the mechanistic effects of anti-TNF-α antibodies in inflammatory bowel disease.

| Antibody | Experimental Samples # | Patients Group | Mechanism of Action | Reference |
|---|---|---|---|---|
| Infliximab | In vitro Jurkat T cell | - | Induction of apoptosis and increase in the Bax/Bcl-2 ration in the CD3/CD28 stimulated cells | [89] |
| | In vitro Jurkat T cells | - | Induction of antibody-dependent cell-mediated cytotoxicity and complement-dependent cytotoxicity | [90] |
| | Ileal epithelium | - | Prevention of the disappearance of occluding-1 and zonula occludens-1 and the increase of claudin-2 tight junction proteins induced by chemical-colitis and TNFα receptor-1 knockout model | [91] |
| | Intestinal epitelial cells * | - | Reduction of intestinal cell apoptosis with reduced expression of membrane bound FAS/CD95 | [92] |
| | Intestinal biopsies and serum * | - | Restoration of epithelial barrier integrity, mucus production and p38-decreased colon inflammation Reduced levels of allograft inflammatory factor-1 in serum and colon | [93] |

**Table 3.** *Cont*.

| Antibody | Experimental Samples [#] | Patients Group | Mechanism of Action | Reference |
|---|---|---|---|---|
| | Intestinal biopsies | HC and CD | Reduction of proliferation marker Ki-67 in endothelial cells, mucosal levels of vascular endothelial growth factor-A, and migration capacity | [94] |
| | Intestinal biopsies | UC and CD | Induction of regulatory macrophages (CD206$^+$/CD68$^+$) in mucosal healing patients | [95] |
| | Intestinal biopsies | HC and CD | Inhibition of granulocyte-macrophage colony-stimulating factor intestinal content, as well as mucosal histology index and peripheral blood leucocyte count | [96] |
| | Intestinal biopsies | CD | Decrease in the immunohistochemical expression of CD31 and vascular endothelial growth factor, correlated to the endoscopic healing | [97] |
| | Intestinal biopsies | HC and CD | Induction of apoptosis of activated lamina propria T lymphocytes, in responding patients | [49] |
| | Intestinal biopsies | CD | Induction of apoptosis in activated T lymphocytes from lamina propria with increase in CD3 and TUNEL positive cells | [89] |
| | LPMC | HC, UC and CD | Induction of cell apoptosis in the co-culture of lamina propria TNFR2$^+$ expressing CD4$^+$ T cells with membrane-bound TNF$^+$ CD14$^+$ intestinal macrophages in the IBD patients | [42] |
| | Intestinal biopsies and PBMC | HC, UC and CD | Increase in CD4$^+$ CD25$^+$Foxp3$^+$ T-regulatory cells and CD4$^+$ CD25 Foxp3$^+$ T-regulatory cells. Decrease of mucosal mRNA and protein expression of Foxp3 in responding patients, but not in non-responders | [98] |
| | Intestinal biopsies and PBMC | HC and CD | Upregulation of IL-22 gene expression in the gut mucosa. Promotion of IL-22 expression by CD4$^+$ T cells through binding to membrane-bound TNF, and Th22 cell differentiation. | [99] |
| | Intestinal biopsies and PBMC | HC, UC and CD | Reduction in the percentage of CD66b$^+$ neutrophils and expression of CD66b in peripheral blood and inflamed mucosa of patients in the response group. Reduction of neutrophil-derived myeloperoxidase, calprotectin, and production of pro-inflammatory mediators, as well as migration of neutrophils | [100] |
| | PBMC | HC and CD | Reduction in the number of circulating monocytes, especially in the classical and intermediate subsets | [101] |
| | PBMC | CD | Lack of influence on the expression of activation markers, homing receptors, memory cells, Fas or Bax/Bcl-2 expression on peripheral blood T lymphocytes | [89] |
| | PBMC | HC and CD | Restoration of the calcium influx and potassium channel function in Th2 lymphocytes comparable to Th1 cells | [102] |
| | PBMC | HC and CD | Restoration of the calcium response and potassium channel function in CD8$^+$ lymphocytes comparable to healthy controls | [103] |

**Table 3.** *Cont.*

| Antibody | Experimental Samples [#] | Patients Group | Mechanism of Action | Reference |
|---|---|---|---|---|
| Adalimumab | In vitro Jurkat T cells | - | Induction of antibody-dependent cell-mediated cytotoxicity and complement-dependent cytotoxicity | [90] |
| | Intestinal biopsies | CD | Decrease in the immunohistochemical expression of CD31 and vascular endothelial growth factor, correlated to the endoscopic healing | [97] |
| | LPMC | HC, UC and CD | Induction of cell apoptosis in the co-culture of lamina propria TNFR2$^+$-expressing CD4$^+$ T cells with membrane-bound TNF$^+$ CD14$^+$ intestinal macrophages in the IBD patients | [42] |
| Golimumab | In vitro Jurkat T cells | - | Binding to transmembrane TNF-$\alpha$ Induction of antibody-dependent cell-mediated cytotoxicity and complement-dependent cytotoxicity | [90] |
| Certolizumab pegol | In vitro Jurkat T cells | - | Binding to transmembrane TNF-$\alpha$ Induction of nonapoptotic cell death in transmembrane TNF-$\alpha$-expressing cells | [90] |
| | LPMC | HC, UC and CD | Induction of cell apoptosis in the co-culture of lamina propria TNFR2$^+$-expressing CD4$^+$ T cells with membrane-bound TNF$^+$ CD14$^+$ intestinal macrophages in the IBD patients | [42] |

[#] Human samples except for those murine studies indicated with an asterisk (*). Abbreviations: CD, Crohn's disease; HC, healthy controls; IBD, inflammatory bowel disease; IL, interleukin; LPMC, lamina propria mononuclear cells; PBMC: peripheral blood mononuclear cells; TNF, tumor necrosis factor; UC, ulcerative colitis.

Antibody-dependent cellular cytotoxicity is an immune mechanism aimed at killing cells marked with antibodies. The Fc region from anti-TNF-$\alpha$ antibodies can bind to Fc receptors from leukocytes and endothelial cells and exerts several cellular functions, including the antibody-dependent cellular cytotoxicity [44]. Hence, recognition of Fc domain by the Fc receptor from effector immune cells, usually natural killer cells, results in cell lysis of the antibody-marked target cell [41]. Likewise, binding of anti-TNF-$\alpha$ to target cells may mediate complement initiation, and activation of its cascade, which results in membrane attack complex, pore formation and subsequent cell death. Induction of both mechanisms were found for infliximab, adalimumab, and golimumab [46,90], but not in the case of certolizumab pegol (which lacks the Fc domain). In relationship with the induction of cell death, anti-TNF-$\alpha$ effects over apoptosis via activation of caspase-family members, modulation of apoptotic markers such as Bcl-2, Bx and Fas, or by transmembrane TNF-$\alpha$ signaling have been shown (Table 3). Nonetheless, conflicting results regarding the action of anti-TNF-$\alpha$ antibodies in modulation of cellular apoptosis have still been reported [44,89,104].

Studies on the regulation of the immune responses including the effects of anti-TNF-$\alpha$ antibodies over cell populations, either activation or repression, as well as over the balance of immune and inflammatory mediators such as cytokines and chemokines, has provided additional insight into the mechanistic effects of these antibodies. Hence, infliximab reduced proliferation marker Ki-67 in endothelial cells [94], the intestinal levels of granulocyte-macrophage colony-stimulating factor [96], the percentage of CD66b$^+$ intestinal neutrophils and expression of blood CD66b [100], and the number of classical and intermediate monocytes subsets [101], while it also induced regulatory CD206$^+$/CD68$^+$ intestinal macrophages [95], and the expression of IL-22 by CD4$^+$ T cells in the gut mucosa [99] (Table 3). Following these immunological studies, infliximab and adalimumab have also been shown to suppress the Fc$\gamma$-receptor-mediated IL-12/IL-23 production by inflammatory macrophages, while certolizumab and etanercept did not [105]. Mucosal IL-33

cytokine, which showed increased levels in acute UC patients, was down-regulated during disease remission by infliximab [106]. Similarly, the chronic and aberrant stimulation of B cells in CD patients was restored in IgM memory B-cell generation and transitional B-cell levels by infliximab [107].

Furthermore, some studies have assessed the local and systemic immunological effects after anti-TNF-$\alpha$ therapy with the use of both intestinal and peripheral samples (Table 3). As an example, the study of Li and colleagues showed that infliximab exerted opposite effects in the Foxp3$^+$ T-regulatory cells in the mucosa and blood, increasing circulating populations while reducing mRNA and protein Foxp3 expression in the gut [98]. This suggested a redistribution of this T-cell subset in IBD. Moreover, the levels of the plasma C-reactive protein inversely correlated with the increase in CD25$^+$ Foxp3$^+$ cells, and durable clinical responses to infliximab were associated with the increase of this circulating cellular subset [98].

Beyond these modulatory changes on cell populations, the impact of anti-TNF-$\alpha$ on cell adhesion molecules and angiogenesis has also been examined [94,97,100], as well as their role in gut barrier function [91,93], and calcium influx and potassium channels [102,103]. These studies provide additional mechanistic basis for anti-TNF-$\alpha$ antibodies, although it should be noted that they have been studied to a lower extent compared with the aforementioned mechanisms [104].

Interestingly, some of the cellular and molecular findings summarized in Table 3 were only or predominantly found in the responding group of IBD patients, in comparison to the non-responders' group by clinical, endoscopic, or histological criteria [44,49,95,97,98,100], thereby linking the mechanistic effects of the anti-TNF-$\alpha$ therapies to their efficacy.

### 3.2. Anti-Integrins Antibodies

Despite the fact that anti-TNF-$\alpha$ antibodies have been predominantly evaluated for their mechanistic effects, some studies have also explored the effects of IBD therapies against trafficking molecules including chemokines and its receptors, integrins and its endothelial ligands, and the molecules acting over lymphocyte recirculation [69]. Table 4 describes some findings that have identified the action mechanisms of anti-integrins antibodies in IBD studies. Indeed, the mechanistic basis of immune cell trafficking and its impact on gut inflammation, homing capacity, and IBD therapies was recently reviewed by Zundler and collaborators [108].

**Table 4.** Summary of studies evaluating the mechanistic effects of anti-integrins antibodies in inflammatory bowel disease.

| Antibody | Experimental Samples | Patients Group | Mechanism of Action | Reference |
|---|---|---|---|---|
| Vedolizumab | Blood | HC | Binding to a subset of peripheral blood memory CD4$^+$ T cells including gut-homing IL-17 T-helper lymphocytes<br>Binding to eosinophils at high levels, and to naïve T-helper lymphocytes, naïve and memory cytotoxic T lymphocytes, B lymphocytes, natural killer cells, and basophils at lower levels<br>The highest level of binding was in the population $\alpha 4\beta 7^{high}$ of memory CD45RO$^+$ CD4$^+$ T cells, specifically in competition with Act-1 | [109] |
| | Intestinal biopsies and PBMC | HC, UC, CD | Blockade of the adhesion of T effector cells from CD patients to MadCAM-1<br>Increase in the expression of intestinal $\alpha 4\beta 1$ integrin in CD | [110] |

**Table 4.** *Cont.*

| Antibody | Experimental Samples | Patients Group | Mechanism of Action | Reference |
|---|---|---|---|---|
| | PBMC | HC, UC, CD | Blockade of the adhesion of peripheral blood leukocytes including CD4+ T cells, CD8+ T cells, CD19+ B cells, and granulocytes to addressin molecules<br>Adhesion was partially, but not completely, related to integrin expression<br>Etrolizumab resulted in similar inhibition of adhesion to MAdCAM-1 than vedolizumab | [111] |
| | PBMC | HC, CD | Higher response of α4β7-expressing lymphocytes to pro-inflammatory cytokines IL-6, IL-7 and IL-21, but lower response to regulatory T cells<br>Enrichment of cells bearing the circulating T follicular helper cell marker CXCR5, in relation with the selective effect of vedolizumab to replace pro-inflammatory effector cells with regulatory T cells and Th2 cells | [112] |
| | Intestinal biopsies and PBMC | HC, IBD (with concomitant HIV-1 infection) | Reduction of intestinal B subsets, naïve and activated CD4+ T cells in the terminal ileum, and lymphoid aggregates within the gastrointestinal tract | [113] |
| | Intestinal biopsies, LPMC and PBMC | HC, UC, CD | Minor effect on lamina propria T cell abundance and mucosal T cell receptor repertoire assessed by immunophenotyping, immunohistochemistry, T cell receptor profiling and RNA sequencing<br>Notable alterations in innate immunity and macrophage populations correlated with the clinical efficacy | [114] |
| | Intestinal biopsies | HC, UC | Partial restoration of the colonic expression of mucosal immune-related genes in UC patients responding to vedolizumab at week 52<br>Significant reduction in the inflammatory cell infiltrate leading to mucosal healing, although persistent histological and gene expression abnormalities remain after therapy in responding patients | [115] |
| Etrolizumab | Intestinal biopsies and PBMC | HC, UC, CD | Blockade of the adhesion of T effector cells from CD patients to VCAM-1 | [110] |
| | Intestinal biopsies and PBMC | HC, UC | Enrichment of pro-inflammatory cytokines, Th17 and Th17/Th1 subsets, and lower expression of regulatory T cell-associated genes in colonic CD4+ T cells expressing higher levels of αEβ7 integrin | [116] |
| Ontamalimab | Blood | CD | Decrease in soluble MAdCAM in serum<br>Increase in β7+ central memory T cells, β7+ effector memory T cells, β7+ naïve T cells, in association with up-regulation of CCR9 gene expression | [117] |

Abbreviations: PBMC, peripheral blood mononuclear cells; LPMC, lamina propria mononuclear cells; HC, healthy controls; UC, ulcerative colitis; CD, Crohn's disease; IBD, inflammatory bowel disease; IL, interleukin; MAdCAM-1, mucosal vascular addressin cell adhesion molecule-1; VCAM-1, vascular cell adhesion molecule-1.

The anti-α4β7 integrin antibody velodizumab blocks the interaction of α4β7 with MAdCAM-1, thereby reducing the gut homing of T cells. Expression of this integrin is high in memory CD4[+] T cells but also in Th2, Th17, B cells, and other immune subsets including innate cells depending on the inflammatory milieu [109,110]. Hence, α4β7 blocking and reduced adhesion to MAdCAM-1 by vedolizumab has been proved in vitro in CD4[+] and CD8[+] T cells, B cells, granulocytes, and monocytes and macrophages [111,118]. Vedolizumab can selectively inhibit the homing of inflammatory T effector cells and decrease lymphoid aggregates in addition to the levels of activated CD38[+] T cells in the ileal region [112,113]. However, one study using a humanized mouse model suggested that CD patients under vedolizumab treatment may display, as a rescue mechanism, increased homing of T effector cells to the ileum via the α4β1-VCAM-1 axis [67].

Following these observations in different immune subsets, a comprehensive analysis of the effects induced by vedolizumab in the mucosal and systemic immunity in IBD patients has been recently performed [114]. Although impairment of lymphocyte homing represents the central mechanism of action, these authors found that vedolizumab exerted minor effects on the abundance or phenotype of intestinal T cells and mucosal T cell receptor repertoire. However, the drug was associated with marked changes in innate immunity, macrophages and molecules involved in microbial sensing, chemoattraction, and the regulation of innate effector responses. Interestingly, this study used infliximab as a comparative control and these effects were not observed in response to the anti-TNF-α antibody [114]. On the contrary, another study has found that the frequencies of circulating innate lymphoid cells, and their alterations in IBD, remained unchanged in patients undergoing vedolizumab treatment [119]. Likewise, treatment with both anti-TNF-α and anti-α4β7 antibodies types had little impact on the profile of circulating B cells subsets in IBD [120].

It is thus suggested that further effects beyond those elicited over the gut homing capacity may arise in future studies [108]. Vedolizumab has been demonstrated to partially restore the colonic expression of immune-inflammatory altered genes in UC patients with endoscopic healing, but only at week 52 and not before [115]. As a suggestive connection, this may be in relationship with the relatively slow onset of the clinical efficacy of this drug and the need to sustain therapy to control the intestinal inflammation [73].

The investigational drug etrolizumab is an anti-β7 integrin antibody and inhibits gut homing through binding of the β7 subunit of both α4β7 and αEβ7 integrins, thereby blocking their interaction with the adhesion ligands MAdCAM-1, VCAM-1, and E-cadherin. As the mechanism of vedolizumab, this leads to internalization of the α4β7 integrin-antibody complex and a diminishing of gut homing [108]. Furthermore, other studies have also shown that the anti-β7 treatment may exert additional mechanisms of action by altering the mucosal retention of pro-inflammatory T cell subsets with higher expression of the heterodimer αEβ7, such as CD8[+] and Th9 cells [110,116]. Indeed, the higher expression of αEβ7 in some immune subsets was suggested to be involved in the more pronounced effect of etrolizumab in comparison to vedolizumab with regards to the reduction of T cell accumulation in the colonic mucosa (Table 4). In agreement with these mechanisms, clinical findings in UC patients under etrolizumab treatment found that clinical remission is higher in patients showing increased tissue basal expression of the αE subunit [83]. Noteworthy, expression of αEβ7 is also relevant in gut CD103[+] dendritic cells, which are altered in IBD and showed impaired ability to generate T regulatory cells [121]. Dendritic cells also govern the tolerogenic/pro-inflammatory profile in intestinal homeostasis. Thus, future studies may address the potential impact of anti-β7-blocking in IBD pathology.

The investigational drug ontamalimab disrupts gut homing via α4β7, but from the endothelial side as an anti-MAdCAM-1 antibody. Treatment with this drug in CD patients provoked changes in the blood cellular composition, increasing β7-expressing T cells and the corresponding CC chemokine receptor 9 (CCR9) gene expression [117]. Moreover, additional effects in the following studies may be expected due to the pleiotropic role

of MAdCAM-1 beyond the α4β7-dependent homing capacity along with its increased expression in extra-intestinal tissues in IBD [82,84].

### 3.3. Anti-Cytokines Antibodies

Cytokines are produced by Th1, Th2, and Th17 cells, monocytes, or intestinal macrophages among others and they have pro- or anti-inflammatory properties. In animal models of colitis, interventions with anti-inflammatory cytokines reduced inflammation [122]. Pro-inflammatory cytokines promote inflammation. Therefore, inhibition of these cytokines may counter their adverse effects in UC and CD. However, until now, there are only anti-TNF antibodies, and those that target the p40 subunit of IL12/23 [122].

IL-12 plays roles in the maintenance of Th1 cells, which secrete, among other molecules, IFN γ [123], while IL-23 plays the same roles for Th17 cells. IL-23 and IL-12 use the same Jak-stat signaling molecules and IL-23R is associated with Jak2 and stat3 [124]. IL-12 induces Th1, which promotes cell-mediated immunity to intracellular pathogens. Animal models and clinical studies indicated abnormal Th1 responses in immune-mediated disorders [125]. On the other hand, IL-23 induces Th17 cells that in humans produce pro-inflammatory cytokines such as IL-17, IL-22, IFN γ, or IL-26 [126].

Ustekinumab inhibits the interaction of IL-12 and IL-23 with their cell surface receptor IL12Rβ1 [127]. This receptor, coupled with IL12Rβ2, forms the IL-12 receptor [124]. The IL12Rβ1 receptor is on the surface of NK cells and T cells. Ustekinumab inhibits the signaling of Il-12 and IL-23, and their activation and cytokine production [77]. Il-12 is formed by two subunits, p40 and p35; the IL12Rβ1 binds to p40 subunit and IL12Rβ2 binds to p35 subunit that is important in intracellular signaling. Therefore, IL-12 may participate in signaling intracellular phosphorylation of STAT4 and STAT6 proteins, NK cell lytic functions, and cytokine production, as mentioned above. IL-23 presents the subunit p19and contains IL-12p40 and uses the IL12Rβ1 for binding to the surface of effector cells. IL-23 also is involved in the intracellular phosphorylation of STAT3, lymphocyte activation, and cytokine production as IL-17A. The specific cellular signaling is due to the association of p19 to the IL-23R [124]. P40 subunit comprises three domains, D1, D2, and D3. The two latest domains are involved in the binding between IL-12p35 and IL-23p19. D1 is the site binding epitope for ustekinumab [128]. Th1 and Th17 cells may induce the production of vasodilators and chemoattractants that promote monocyte and neutrophil recruitment, T cell infiltration and neovascularization [77].

Ustekinumab neutralizes both IL-12 and IL-23 and has a specificity for the subunit p40 of human IL-12 and IL-23. This antibody binds the D1 domain of IL-12/IL-23 p40 with electrostatic interactions and shape complementarity that stabilizes the ustekinumab/p40 interface. Curiously, ustekinumab does not bind mouse IL-12 and IL-23. The mouse subunit p40 has three amino acids residue clusters different from the human p40 [128]. Lu et al. reported the crystal structure of the IL-12/ustekinumab Fab complex revealing that this monoclonal antibody recognizes an epitope on D1 domain of the p40 subunit. In fact, mutations of residues in the epitope showed a reduction in ustekinumab binding. The epitope is spatially distant from IL-12p35 and lL-23p19 and is available in both IL-12 and IL-23. Therefore, there is a dual neutralization of both cytokines mediated through p40 due to the equal binding of ustekinumab. The lack of hydrophobic-hydrophobic contacts in the ustekinumab/p40 binding interface suggests that other kind of interactions are predominant; there are electrostatic interactions as salt bridges that stabilize the proteins in this case [128]. The crystal structures of the bound Fab ustekinumab and the free Fab ustekinumab showed that antigen binding does not induce changes in the structure. Therefore, the antigen binding site is rigid and is optimized for recognition. The molecular structures of the signaling for IL-12 and IL-23 has not been elucidated yet. It is likely that ustekinumab epitope is close to the interaction site between IL12Rβ1 and IL-12p40 [128].

Table 5 collects all of these studies about p40 subunit.

**Table 5.** Action mechanisms of ustekinumab and IL-12/23.

| Experimental Techniques | Mechanism of Action | Reference |
|---|---|---|
| Cell lines: hIL-2 dependent T cell line kit225 (DNAX, Palo Alto, CA); mIL-3 dependent Ba/F3 cells; NKL cells Signal transduction: SDS-PAGE and immunoprecipitation. Western-blot | IL-12 and IL-23 have similar signal transduction mechanisms such as Jak2, Tyk2, Stat1, Stat3, Stat4, Stat5 IL-12 participates in phosphorylation of Stat4 and Stat 6, NK cell lytic functions IL-23 participates in phosphorylation of Stat3 and lymphocyte activation IL-12 and IL-23 produce pro-inflammatory cytokines. | [77,124] |
| Human Ig transgenic mouse technology | Identification of monoclonal hybridoma clone that produces huma IgG that binds and neutralized IL-12: ustekinumab Description of IL-23 was later and due to the discovery of ustekinumab | [77] |
| Hu-Ig mice technology | Binds to p40 of IL-12 and IL-23 preventing their interaction with IL-12Rβ | [77,127] |
| Crystal structure studies | D1 of p40 binds epitope for ustekinumab | [77,128] |
| Isothermal titration colorimetry analysis | Ustekinumab binds to IL-12 and IL-23 equally | [77,128] |

Abbreviations: hIL-2, human interleukin 2; mIL-3, mouse interleukin 3; NKL, natural killer cell line; Ba/F3, IL-3 dependent murine pro B cell line; SDS-PAGE, sodium dodecyl sulfate polyacrylamide gel electrophoresis; IL, interleukin; Jak2, Janus kinase 2; Tyk2, tyrosine kinase 2; Stat, signal transducer and activator of transcription; NK, natural killer; Ig, immunoglobulin; Hu-Ig, Human immunoglobulin; IL-12Rβ, interleukin 12 receptor β.

## 4. Conclusions

Nowadays, there are many available treatments for IBD, from conventional to biological or small molecules.

Biological treatments are very successful in the therapy of IBD. However, these treatments are still expensive and new patients with IBD must begin first with the traditional treatments without knowing if they will work for them. On the other hand, IBD has no cure, and even with these novel treatments, patients must frequently switch their medication and undergo colonoscopy. Moreover, many patients do not respond correctly to treatments and frequently surgery is their only option. Therefore, new treatments (both biological and small molecules) are constantly being tested.

Despite the efforts made in recent years to fill the gap in the mechanistic knowledge of biologicals, particularly regarding anti-TNF-α therapies, further studies are needed in order to better understand the action mechanism of these drugs, which will help understand how to improve efficacy and safety. These studies will hopefully pave the path to a personalized medicine.

**Author Contributions:** Conceptualization: R.A.; writing—original draft preparation, L.O.M., S.F.-T. and R.A.; writing—review and editing, R.A. All authors have read and agreed to the published version of the manuscript.

**Funding:** This research was funded by Ministerio de Ciencia, Innovación y Universidades, grant number PID2019-111510RB-I00.

**Data Availability Statement:** No new data were created or analyzed in this study. Data sharing is not applicable to this article.

**Acknowledgments:** L.O.M. has a fellowship from Comunidad de Madrid and Universidad Autónoma de Madrid (Ayudas Atracción de Talento modalidad 2. BMD-5800); S.F.T. had a fellowship from Instituto de Salud Carlos III (Sara Borrell CD17/00014).

**Conflicts of Interest:** The authors declare no conflict of interest.

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
