# Peer review of "Biological Treatments in Inflammatory Bowel Disease: A Complex Mix of Mechanisms and Actions"

_biologics, doi:10.3390/biologics1020012_

Round 1

Reviewer 1 Report

Dear Authors,

I believe that your manuscript lacks in novelty and major revisions should be done in order to be published. Many of the covered subjects have been previously analyzed by others, but most importantly, your manuscript requires significant editing due to the observed plagiarism.

Kind regards,

The reviewer

Reviewer 2 Report

This is a review speaking of mechanism of action of the major biological therapies used for IBD. I think that, although it is an interesting topic, the paper does not provide any new information, nor gives some practical advice for clinical practice. Given that, mechanism of action of bioloogical therapies is a fascinating topic so the review is woth publushing but after some rivisions.

Most of all Table 1 needs some editing. The classification based on localisation is not very precise: Ulcerative colitis divides into proctitis, left sided colitis and pancolitis (or extensive colitis) which does not need to involve the whole colon but is, biy definition, any colitis thas passes spelic flexure. In the clinical features anemia is part of both the dieasese (also in UC because of the bleeding). In addition pathogenesis may base also on dysbiosis but, at the moment, it is not so clear, some authors believe that dysbiosis is a consequence of the inflammation. In the enbd, the mosts important part that needs extensive revision (or even elimination) is the part regarding diet. There is no clear evidence about influence of diet in IBD flares. In addition, certainly high fiber diet (fruits and vegetables) is not recommended in Crohn's disease especially if stricturing.

Reviewer 3 Report

The aims of this manuscript are to review current biological treatments for IBD and discuss about mechanisms of action in anti-TNFα antibodies and anti-integrin agents. It appears to be interesting topics. Especially, the mechanistic insight into anti-TNFα agents looks great. Therefore, the authors may need to expand/add new discussion regarding mechanisms of action for the other drugs as well (section 4 and new 5…). Several comments/suggestions are shown below.

  1. Tables need to be modified. It is difficult to understand the contents quickly in this format, particularly Table 1.
  2. References for each description in the table 1 should be shown as some has a lack of sufficient evidence or still under debate. For examples, regarding “pathogenesis”, not only Th2 but also other types of immune dysfunction are involved in the pathogenesis of UC.
  3. The authors discuss the mechanism of action in anti-TNFα in depth, while the mechanism of neither anti-integrin agents nor anti-IL-12p40 drug is discussed well. It might be better to focus on just anti-TNFα agents or expand/add new discussion regarding mechanisms of action for these biological drugs.

Round 2

Reviewer 1 Report

Dear authors,

I would like to thank you for your effort and the new additions to the texts.

I have no other comments to make.

Reviewer 3 Report

The revised manuscript now contains precise description about the mechanisms of action in anti-integrin and anti-IL-12p40 agents. Tables have also been modified. However, the revised version of Table 1 contains just general statements for IBD but not any novel information. I rather recommend removing it. Instead, tables summarized studies evaluating the mechanisms of action in anti-integrin and ant-IL-12p40 agents (like Table 3 and 4 for anti-TNF-α antibodies) would be more interesting and fit this review manuscript regarding biological treatment for IBD.
